

# Responses of fossil coccolith morphology to preservation conditions in the deep ocean

Authors: Amanda Gerotto[1,3]*, Hongrui Zhang[2]*, Renata Hanae Nagai[3], Heather M. Stoll[2], Rubens César Lopes Figueira[1], Chuanlian Liu[4], Iván Hernández-Almeida[2]

[1]Oceanographic Institute, University of São Paulo, São Paulo, Brazil; [2]Geological Institute, ETH Zurich, Zurich, Switzerland; [3]Center for Marine Studies, Federal University of Paraná, Pontal do Paraná, Brazil; [4]State Key Laboratory of Marine Geology, Tongji University, Shanghai, China.

*Corresponding authors: gerottoamanda@alumni.usp.br, zhh@ethz.ch

## Abstract

Understanding the variations in past ocean carbonate chemistry is critical in elucidating the role of the oceans in balancing the global carbon cycle. The fossil shells from marine calcifiers present in the sedimentary record are widely applied as past ocean carbon cycle proxies. However, the interpretation of these records can be challenging due to the complexity physiological and ecological response to the carbonate system during organisms' life cycle, as well as the potential for preservation at the sea-floor. Here we present a new dissolution proxy based on the morphological attributes of coccolithophores from the Noëlaerhabdaceae family (*Emiliania huxleyi* and *Gephyrocapsa* spp., > 2 μm). To evaluate the influences of coccolithophore calcification and coccolith preservation on fossil morphology, we measured morphological attributes, mass, length, thickness, and shape factor (ks), of coccoliths in a laboratory dissolution experiment and surface sediment samples in the South China Sea. The coccolith morphological data in surface sediment were also analyzed with environment settings, namely surface temperature, nutrients, pH, chlorophyll-a concentration, and carbonate saturation of bottom water by a redundancy analysis. Statistical analysis indicate that carbonate saturation of the deep ocean explains the highest proportion of variation in the morphological data instead of the environmental variables of the surface ocean. Moreover, the dissolution trajectory in the ks vs length of coccoliths is comparable between natural samples and laboratory dissolution experiments, emphasizing the importance of carbonate saturation on fossil coccolith morphology. However, the mean ks alone cannot fully explain all variations observed in our work. We propose that the mean ks and standard deviation of ks (σ) over the mean ks (σ/ks) could reflect different degrees of dissolution and size-selective dissolution, influenced by the assemblage composition. By applying together with the σ/ks ratio, the ks factor of fossil coccoliths in deep ocean sediments could be a potential proxy for a quantitative reconstruction of past carbonate dissolution dynamics.



35

## 1. Introduction

The ocean's large reservoir capacity of carbon dioxide ($CO_2$) plays an essential role in the carbon cycle and, consequently, in controlling atmospheric $CO_2$ (Ridgwell and Zeebe, 2005; Wang et al., 2016). The ocean $pCO_2$ is influenced by temperature, salinity, and biological activity, including primary production, respiration, calcification, and carbonate dissolution (Ridgwell and Zeebe, 2005; Sarmiento and Gruber, 2006; Libes, 2009; Wang et al., 2016). When $CO_2$ dissolves in water, the ocean becomes more acidic, decreasing pH, carbonate ion concentration, carbonate saturation ($\Omega_{Ca}$). The carbonate compensation depth (CCD) is the depth at which the rate of calcite dissolution is balanced by the rate of calcite supply. The CCD is usually several hundred meters deeper than the chemical lysocline, the saturation horizon of calcite, due to the kinetics of dissolution (Ridgwell and Zeebe, 2005). Whereas the photic zone is supersaturated with respect to calcite in most areas of the ocean, large areas of the deep ocean are currently undersaturated because of the increased solubility of calcite with pressure (Sulpis et al., 2018). As the ocean continues absorbing larger amounts of $CO_2$ from anthropogenic fuel emissions, a shallowing of the CCD is expected for the next 100 years due to the sharp decrease of carbonate saturation from surface to deep ocean (Hönisch et al., 2012; USGCRP, 2017; Sulpis et al., 2018; IPCC, 2019). Variations in the CCD on timescales from millions to several thousands of years are an important process in determining the ocean's carbonate chemistry and regulating atmospheric $CO_2$ (Emerson and Archer, 1990; Pälike et al., 2006). Understanding the role of physical and biogeochemical parameters in marine carbonates is therefore critical to interpret the geological record correctly and to reconstruct variations of the ocean carbon cycle in the past.

The effects of carbonate chemistry changes and variations in the position of the CCD in the geological past have been investigated using a wide array of geochemical and microfossil proxies such as $\delta^{13}C$ in benthic and planktonic foraminifera (Zachos et al., 2005; Hönisch et al., 2012), fragmentation indices of calcareous microfossils (Le and Shackleton, 1992; Broerse et al., 2000; Flores et al., 2003), and $CaCO_3$ content (Archer et al., 2000; Palike et al., 2006) in marine sediments. However, these proxies do not provide quantitative estimates of past changes in carbonate chemistry because some additional ecological mechanisms determine the calcification and preservation responses (Hönisch et al., 2012; Rae et al., 2021). $\delta^{11}B$ provides quantitative proxy for past seawater pH (Hönisch et al., 2012), albeit additional carbonate chemistry parameters impose some limits on the interpretation of the proxy (Yu and Elderfield, 2007; Rae et al., 2021). Benthic B/Ca provides a quantitative proxy for deep sea $CO_3^{-2}$





concentration (Yu et al., 2016). Yet both of these methods require mono-specific foraminifera
samples for measurements, which are time-consuming to pick, and analyses are limited to
sediment samples that contain sufficient concentration of this microfossil group.

Coccolithophores, a group of single-celled calcifying algae, are characterized by the

production of calcite plates called coccolith. Coccoliths are the main constituent of marine
biogenic sediments, contributing up to 80 % to deep-sea carbonate fluxes (Young and Ziveri,
2000; Hay, 2004). Changes in coccoliths morphology were believed to record the evolution
history of coccolithophores and reflect the environmental conditions in the surface ocean (i.e.
during coccolith biomineralization) (Riebesell et al., 2000; Iglesias-Rodriguez 2008; Beaufort et
al., 2011; Charalampopoulou et al., 2016; Rigual-Hernández et al., 2020a). Because of that,
coccoliths are widely used in paleoclimate and paleoceanographic reconstructions (e.g., Rickaby
et al., 2007; Henderiks and Pagani, 2007; Bolton et al., 2016; Bollman and Herrle, 2007). Several
methods exist to estimate coccolithophore calcification in the fossil record. Assumed
proportional length and thickness allowed for the first estimates of coccolith mass using
microscope techniques (Young and Ziveri, 2000). More recent methods based on the optical
properties of calcite under polarized light microscopy (circular and linear) allowed a more
precise estimate of the thickness of individual coccoliths (Beaufort, 2005; Beaufort et al., 2021;
Bollman et al., 2014; Fuertes et al., 2014; Johnsen and Bollmann, 2020). The optical techniques
have been successfully employed in downcore records to estimate coccolithophore calcification
across time and evolutionary steps (e.g., Bolton et al., 2016; Beaufort et al., 2022; Guitián et al.,
2022). However, until now there is no study that evaluates the response of calcification patterns
of fossil coccolithophores to both environmental parameters controlling biomineralization in the
photic zone and calcite saturation state at the depth of burial.

The South China Sea (SCS) is the largest marginal basin of the Western Pacific,

characterized by very dynamic spatial environmental conditions and a steep bathymetric profile
(Wang et al., 2015). Sediment records from this basin have been used to study the response of
coccolithophores to different environmental variables. Previous studies found positive
correlations between coccolithophores biometry from plankton samples and nutrients and light
at the photic zone (Jin et al., 2016). Building up on these results, but applied to the sedimentary
record, Su et al. (2020) explored the dependency of coccolithophore weight and past surface
ocean carbon chemistry parameters and nutrient conditions. However, it has been also
demonstrated that there is intense coccolithophore dissolution above the lysocline in the SCS
(Fernando et al., 2007a). More recently, a study using plankton tow material found that the
degree of calcification in the coccolithophore species *Emiliania huxleyi* was insensitive to
carbonate chemistry in surface waters (Jin et al. 2022a). This diversity of results calls for new



studies that explore systematically the drivers of coccolithophore morphology and calcification
in the fossil record.

Here, we analyzed morphological attributes of fossil coccolithophores in surface

sediment samples (n = 28) in the SCS, located across spatial environmental gradients in the
surface ocean, but also across a bathymetric transect related to the calcite saturation at the sea
floor which leads to lower calcite saturation at the sea-floor in deeper sites. In addition, we
evaluated the morphological variations of coccoliths under different dissolution intensities in a
laboratory experiment. Using an automated algorithm to estimate coccolithophore calcification
from images taken with a microscope under cross-polarization, we show that scale-invariant
measures of coccolith thickness (shape factor, ks) from coccolithophores located along a depth
gradient in the SCS are highly correlated to the calcite saturation state at the seafloor. We
propose a new calibration to reconstruct past calcite saturation based on ks which would enable
the quantitative reconstruction of changes in the calcite saturation in the deep ocean and
position of the CCD in the past.

## 2. Oceanographic settings

The SCS is a marginal basin located in the Western Pacific, connected to the open ocean

by north and south shallow passages (Fig. 1A). The Luzon Strait in the north is the deepest (~2000
m) and the principal channel for water exchanges between the SCS and the Pacific through the
Kuroshio Current (Qu et al., 2006; Liu et al., 2011; Wan and Jian, 2014).  The modern surface
circulation and hydrographic characteristics of the SCS are directly associated with the seasonal
changes promoted by the East Asian Monsoon (Wang and Li, 2009). These seasonal
hydrodynamic patterns control the regional sea surface temperature (SST) distribution, salinity,
and nutrients (Fig. 1B-E, Wang and Li, 2009). The SST latitudinal gradient is up to 2° C with an
annual average of 28-29° C in the southern SCS and 26-27° C in the north (Tian et al., 2010).
Salinity varies seasonally between 32.8-34.2 psu, with smaller salinity variation in the north than
in the south (Wang and Li, 2009). Northern SCS primary productivity reflects the seasonality of
the EAM with more productive and well-mixed waters during the winter season (Zhang et al.,
2016), with higher chlorophyll-$a$ concentration (0.65 mg Chl-α m$^{-3}$ and 600 mg C m$^{-2}$ d$^{-1}$) (Chen,
2005; Chen et al., 2006; Jin et al., 2016).




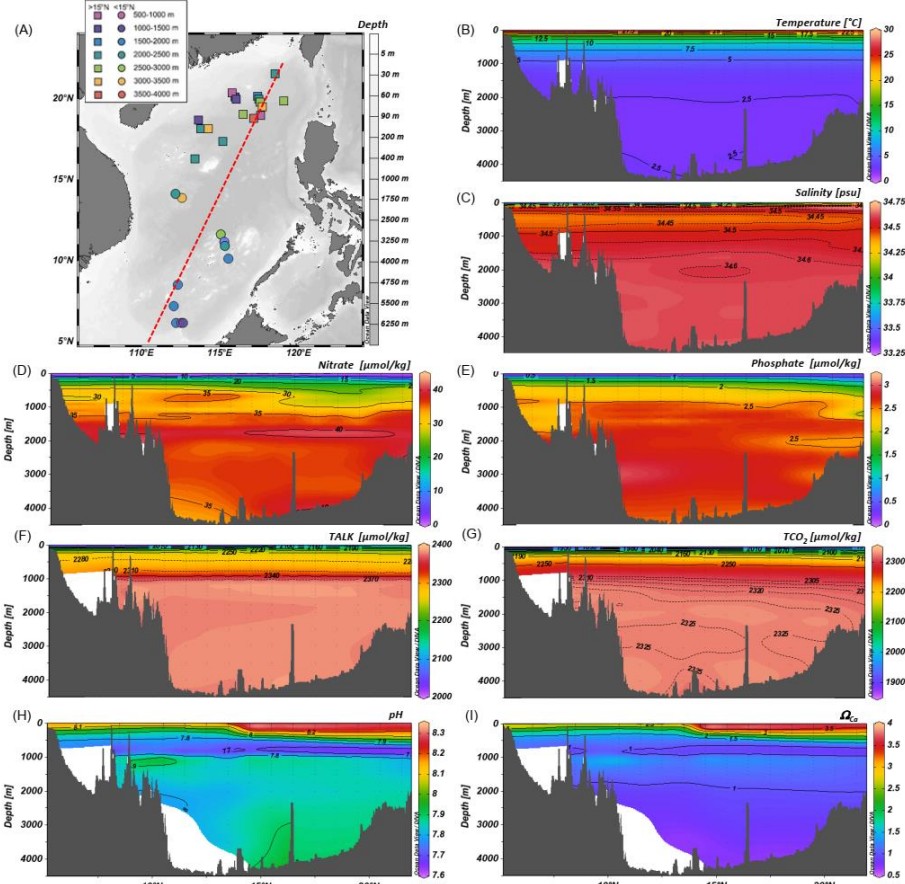


Figure 1. Map of the South China Sea and location of core-top samples used in the present study. Dots and squares represent stations located from 6° to 15° N, and from 15° to 22° N. (A). Vertical profiles along N-S (5° to 22° N) transect (dotted red line on panel A) of (B) temperature, (C) salinity, (D) nitrate, and (E) phosphate obtained from World Ocean Atlas 2001 (WOA01, Conkright et al., 2002), (F) total alkalinity (TALK) and (G) total inorganic carbon concentration (TCO$_2$) from Goyet et al. (2000), (H) pH and (I) $\Omega_{Ca}$ calculated at CO2SYS (Pierrot et al., 2012) from previously extracted data (Goyet et al., 2000). The map and the vertical profiles were plotted with ODV software (Schlitzer, 2019).

The modern SCS lysocline is approximately 1200 m, and the CCD lies between 3500 and 3800 m (Thunell et al., 1992; Wang et al., 1995; Luo et al., 2018). In the northern SCS, surface waters (e.g., the upper 300 m) are characterized by relatively lower DIC and TALK (Fig. 1F-G) and





higher pH and $\Omega_{Ca}$, compared to deeper waters (Fig. 1H-I) (Chou et al., 2007; Jin et al., 2016).
Below 1000 m, the SCS across a N-S transect is characterized by relatively homogeneous DIC,
$\delta^{13}C$, and $[CO_3^{2-}]$ (Chen et al., 2006; Qu et al., 2006; Chou et al., 2007; Wan et al., 2020).
The SCS deep waters originate from the North Pacific Deep Water (NPDW) that
penetrates the marginal basin through the Luzon Strait (Qu et al., 2006; Liu et al., 2011; Wan
and Jian, 2014; Wan et al., 2018). The route traced from the Luzon Strait to the northwest
suggests a predominantly cyclonic deep circulation (Qu et al., 2006; Wang and Li, 2009). The
deep-water residence time of the SCS is estimated to be approximately 30-50 years, like that of
intermediate waters, 52 years (Chen et al., 2001). Due to this short residence time, the SCS
presents a homogeneous vertical profile; below 2000 m, there are no evident chemical
stratification or changes compared to the Pacific deep-water chemistry (> 2000 m)
characteristics (Chen et al., 2001; 2006; Qu et al., 2009). The rapid residence time potentially
implies that, when replaced, deep waters occupy intermediate water levels (between 300 m and
1300 m), contributing to the circulation of intermediate and shallow waters and ocean-
atmosphere exchanges (Qu et al., 2009; Tian et al., 2010).

## 3. Material and methods

### 3.1 Material and sample treatments

The core-top samples (n =28) employed in this study were retrieved from different
depths in the basin of SCS (Fig. 1) during the R/V Sonne cruises (SO-95) (Table 1). Toothpick
samples from each location were used to prepare smear slides, without any chemical or physical
treatment. Unfortunately, the surface sediments were already depleted resulting in not having
enough material to perform dissolution experiment using the same samples. For the dissolution
experiment, we employed 240 mg of dry sediment obtained from the Late Pleistocene sample
from the Western Equatorial Pacific (ODP 807A-2H-2W, 57-59 cm). This sample contains a higher
abundance of the species *Gephyrocapsa caribbeanica* compared to the thinner species found in
the SCS (Roth and Coulbourn, 1982; Roth and Berger, 1975). The sediment sample was
suspended in 120 ml Milli-Q water, and then the suspension was evenly separated into 6
centrifuge tubes, each a volume of 20 mL and containing the equivalent of 40 mg of sediment.
Sodium hexametaphosphate $(NaPO_3)_6$ (Calgon®) has been traditionally used in pretreatment of
samples with calcareous microfossils, particularly foraminifera (Olson and Smart 2004; Smart et
al., 2008). However, it has been observed that application of this chemical agent dissolves these
microfossils due to complexation of Ca with phosphates, an effect which varies with the
exposure time (Feldmeijer et al., 2013). Therefore, we added 100 mg of Calgon® into 100 ml
Milli-Q water, resulting in a concentration of 1.6mM, to conduct our dissolution experiment.



Different volumes of Calgon® solution (0, 0.4, 0.8, 2, 4 6 ml) were added to each of the six
subsamples. The Calgon® is very corrosive to the fine carbonate particles, and the reaction
between Calgon® and carbonate could be simplified in two steps. First, the $(NaPO_3)_6$ hydrolysis
releases the sodium trimetaphosphate $(Na_3P_3O_9)$. Then, the calcium in the solution is exchanged
with sodium and precipitate as $Ca(PO_3)_2$, $CaNa(PO_3)_3$, and $CaNa_4(PO_3)_6$, strongly reducing the free
calcium concentration in the solution. The decrease in calcium concentration promotes
carbonate dissolution. In theory, adding 1 mol $(NaPO_3)_6$ would result in the dissolution of 3 mol
$CaCO_3$ at maximum. So, there could be ~80 % carbonate left even after adding 6 ml Calgon®
solution. The particles in all tubes were kept suspended gently by a rotating disaggregation
wheel as described previously (Stoll and Ziveri, 2002) for two days to achieve a full reaction
between carbonate and $(NaPO_3)_6$. Slides were prepared for coccolith morphological analyses
using the drop technique as described by Bordiga et al. (2015) to trace the variations of coccolith
amount during dissolution.

3.2 Coccolith morphological parameters

The morphological parameters of coccolith in the dissolution experiment and surface

sediment were analyzed using a Polarized Microscope (Zeiss Axio Scope HAL100), configured
with circularly polarized light and a Zeiss Plan-APOCHROMAT 100x/1.4 oil objective, and a
coupled camera (Zeiss Axiocam 506 Color). For every sample, at least 40 fields of view were
photographed. After species identification and selection of coccolithophores images belonging
to the Noëlaerhabdaceae family (*Emiliania huxleyi* and *Gephyrocapsa* spp, > 2 μm), each sample
had between 100 and 400 (average of 250 per sample) coccolithophore images. The relationship
between the color of coccolith images and thickness was calibrated using a reference calcite
wedge, the thickness of which had been carefully quantified (González-Lemos et al., 2018). After
calibration, all images were analyzed in the Matlab-based software, C-Calcita (Fuertes et al.,
2014), to obtain the coccolith morphological parameters, including length, volume, and mass.
The length-shape factor of each coccolith, ks, was calculated using the formulation by Young and
Ziveri (2000) based on the coccolith mass and length obtained using C-Calcita:
$ks = \frac{Mass}{2.7 \times Length^3}$

Beyond the traditional morphological parameters, we calculated the ratio of the

standard deviation of ks (σ) over the mean ks (σ/ks). The goal of this novel parameter is to
provide a new dimension to trace the dissolution process in coccoliths, especially when the
coccolith assemblage is diverse. For example, if the coccoliths dissolve at different speeds in the



assemblage due to differential sensitivity to acidification, a small increase of σ/ks would be
expected at the beginning of the dissolution because of the σ increase and ks decrease. Then,
after all fragile coccoliths dissolve, leaving only thicker coccoliths in the assemblage, the σ/ks
should show a decreasing trend which could be mainly caused by a decrease σ.

Table 1. Station, coordinate data, and water depth of core-top samples used in this study.

| Station | Longitude (E) | Latitude (N) | Water depth (m) |
|---|---|---|---|
| 17930 | 115.782 | 20.333 | 629 |
| 17965 | 112.552 | 6.157 | 889 |
| 17943 | 117.553 | 18.95 | 917 |
| 17931 | 115.963 | 20.1 | 1005 |
| 17944 | 113.637 | 18.658 | 1219 |
| 17963 | 112.667 | 6.167 | 1233 |
| 17932 | 116.037 | 19.952 | 1365 |
| 17964 | 112.213 | 6.158 | 1556 |
| 17960 | 115.558 | 10.12 | 1707 |
| 17940 | 117.383 | 20.117 | 1728 |
| 17961 | 112.332 | 8.507 | 1795 |
| 17959 | 115.287 | 11.138 | 1957 |
| 17962 | 112.082 | 7.182 | 1970 |
| 17949 | 115.167 | 17.348 | 2195 |
| 17957 | 115.31 | 10.9 | 2197 |
| 17941 | 118.483 | 21.517 | 2201 |
| 17951 | 113.41 | 16.288 | 2340 |
| 17945 | 113.777 | 18.127 | 2404 |
| 17955 | 112.177 | 14.122 | 2404 |
| 17939 | 117.455 | 19.97 | 2473 |
| 17958 | 115.082 | 11.622 | 2581 |
| 17934 | 116.462 | 19.032 | 2665 |
| 17938 | 117.538 | 19.787 | 2835 |
| 17925 | 119.047 | 19.853 | 2980 |
| 17956 | 112.588 | 13.848 | 3387 |
| 17937 | 117.665 | 19.5 | 3428 |
| 17946 | 114.25 | 18.125 | 3465 |



| 17936 | 117.12 | 18.767 | 3809 |
|---|---|---|---|



### 3.3 Environmental data for surface sediment

Annual means of different physical, chemical and biological variables in both 50 m depth
and bottom water for the location of the surface samples (Table 1) were extracted from different
databases. Here the 50 m depth was selected because it is the depth at which the highest
concentration of Noëlaerhabdaceae coccolithophorid has been observed in the SCS (Jin et al.,
2016). Seawater temperature, salinity, phosphate, and nitrate concentrations at 50 m were
obtained from WOA01. Sea surface chlorophyll-a concentration data were based on MODIS data
(2003-2016) extracted from Oregon State University Ocean Productivity
(http://www.science.oregonstate.edu/ocean.productivity/). Annual averaged concentrations of
total alkalinity (TALK) and dissolved inorganic carbon ($TCO_2$) were extracted from Goyet et al.
(2000). Then the carbonate ion concentration, pH, $pCO_2$ for the depth of 50 m and $\Omega_{Ca}$ for the
sea floor depth were calculated by CO2SYS macro for Excel® (Pierrot et al., 2012) using extracted
variables, salinity, temperature, pressure, total phosphate, total silicate, TALK, and $TCO_2$ at the
corresponding depth (50 m or depth of the surface sediment sample). The light intensity at 50
m water depth was calculated using a model of penetration of photosynthetic active radiation
(PAR) from surface to depth (Buiteveld, 1995; Murtugudde et al., 2002), monthly climatologies
of PAR from the MODIS Ocean database (http://oceancolor.gsfc.nasa.gov/cgi/l3), and the
diffuse attenuation coefficient for downwelling irradiance at 490 nm (Kd490) and Equation 1 in
Lin et al. (2016).

### 3.4 Statistical analysis

Pearson correlation and redundancy analysis (RDA) were employed to explore the
relationship between morphological features of the coccoliths in surface sediment samples and
the environmental data. All statistical analyses were performed using the PAST 4.06 software
(Hammer et al., 2001).

## 4. Results

### 4.1 Variations of coccoliths morphology in the dissolution experiment

In the dissolution experiment, mean ks decreased with increasing volume of Calgon®
solution (Fig. 2A). The mean ks varied between 0.12 (0 ml Calgon®) and 0.04 (6 ml Calgon®) (Fig.
2A). The σ/ks represents variation in preservation among coccoliths within each sample. Higher



differences in σ were observed in samples containing 2 ml, 4 ml, 0.8 ml, 0.4 ml, 0 ml, and 6 ml,
respectively (Fig. 2B). Increasing the amount of Calgon® solution up to 2 ml showed a decrease
in mean ks and an increase in σ. Samples with 4 and 6 ml Calgon® solution showed a reduction
in mean ks and σ among coccoliths (Fig. 2B). The lowest mean ks (0.04) and the maximum mean
length (3.95 μm) were recorded under the higher Calgon® solution (6 ml) amount (Fig. 2C).
Increased amounts of Calgon® solution also resulted in a gradual increase in coccolith length
leading to a negative correlation between length and ks (r= -0.68, p > 0.05), but not significant
due to the small number of observations (Fig. 2C).

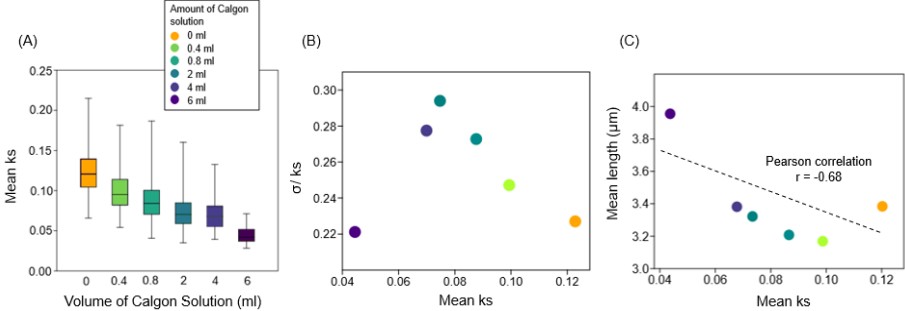


Figure 2. Coccolith morphological variations in the dissolution experiment. (A) Box plots of the
median (horizontal line inside the boxes), minimal and maximal values of coccoliths mean ks
(vertical bars) under the different volumes of Calgon® solution; (B) Scatter plot of mean ks and
σ/ks and (C) linear correlation and correlation coefficient (p > 0.05) between mean ks and mean
length.

4.2 Biological and environmental effects on coccolith morphology

Overall, the mean ks, thickness, and volume in the core-top sampling stations (Fig. 3)

presented higher values in shallower depths. The mean ks varied between 0.03 and 0.07, and
the mean thickness was between 0.25 and 0.44 μm, with maximum values recorded at station
17931 located in northern SCS at 1005 m water depth. The mean length of coccolith varied
between 3.23 and 3.78 μm, with the highest values recorded at 2195 m water depth (site 17949)
in northern SCS, but without a significant trend along depths. The mean volume of coccoliths
ranged between 1.70 and 2.97 μm$^3$, and the mean mass was between 4.61 and 8.03 pg, with
maximum values for both recorded in the shallowest station, 17930, at 629 m water depth in
northern SCS.



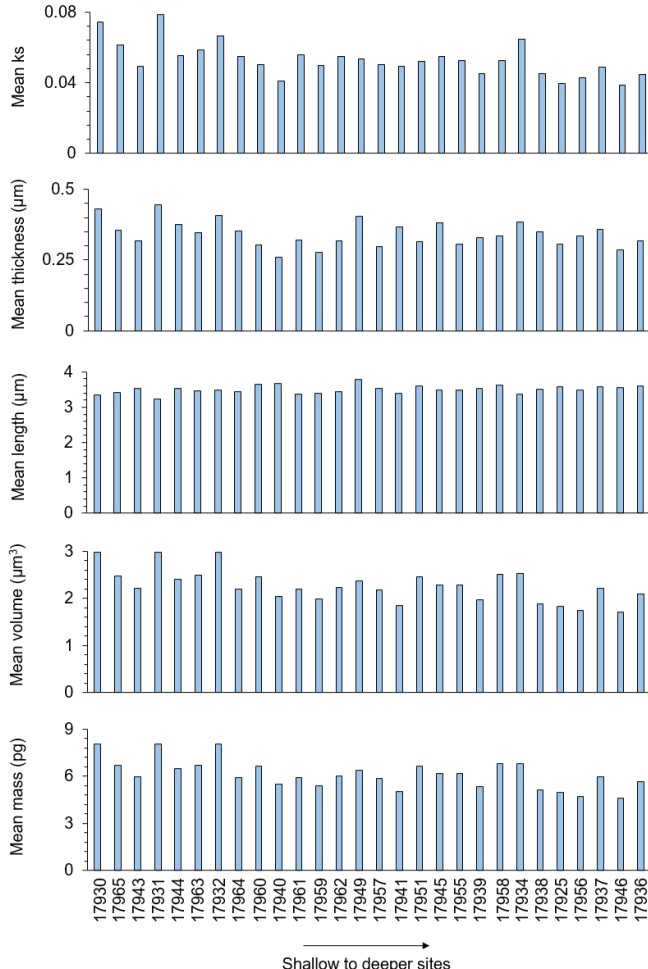

Figure 3. Coccolith mean ks, thickness (μm), length (μm), volume (μm³), and mass (pg) in surface samples from SCS. The sampling stations are distributed along the x-axis according to their depth, sorted from the shallowest to the deepest.

In general, the degree of dissolution varied according to the depth of the sediment samples. The σ/ks vs. ks presents different trajectories associated with light, strong, or no dissolution (Fig. 4A). The shallowest stations south (<15˚ N) and north SCS (> 15˚ N) show a linear and increasing trend between ks and σ/ks. For the samples below 2000 m, there is no clear pattern of variation related to the mean ks standard deviation. However, samples below 3000 m are mainly located on the left upper part of the plot, in a similar position as the samples treated with 4 and 6 ml of Calgon® in the ks vs. σ/ks comparison of the dissolution experiment



(Fig. 3B). The mean ks vs. mean length shows a negative correlation (r = -0.62, p < 0.05), with
the deepest samples showing larger size coccoliths and lower mean ks (Fig. 4B).


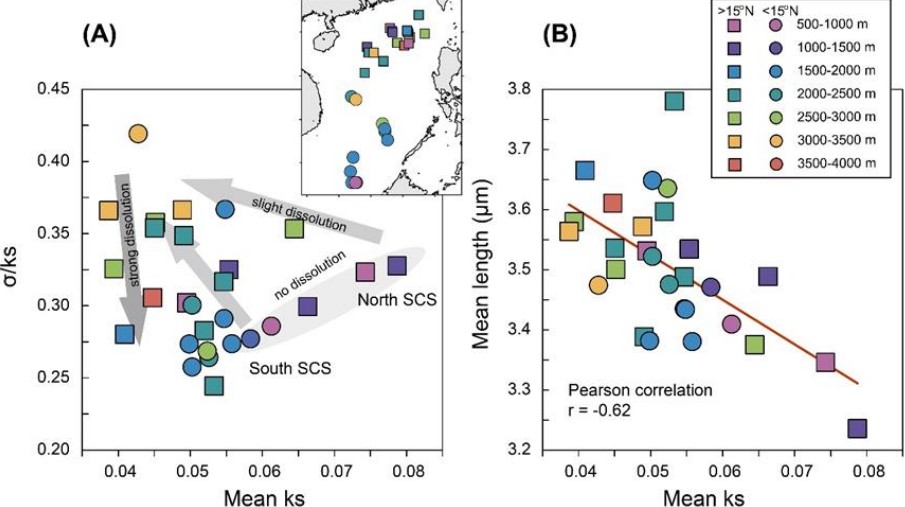


Figure 4. Morphological parameters of coccolith in surface sediments (A) Scatter plot between
mean ks and σ/ks and (B) linear correlation and correlation coefficient (p < 0.05) between mean
ks and mean length. Shaded arrows in panel A represent ideal trajectories of mean ks vs σ/ks as
shown in figure 2B, to interpret the trends in the surface sediment samples. Note that the mean
ks of figures 2 and 4 are different due to the higher abundance of the species *Gephyrocapsa*
*caribbeanica,* with higher thickness, in the sample for the dissolution experiment.

We analyzed the correlations between the biological and environmental datasets (Table
2). Significant correlations can be found (p < 0.05) between several morphological parameters
of coccolith and bottom water carbonate chemistry ($\Omega_{Ca}$), with a correlation coefficient (r) = 0.67
between mean ks and $\Omega_{Ca}$, r = 0.66 between mean volume and $\Omega_{Ca,}$ and r = 0.66 between mass
and $\Omega_{Ca.}$ The mean thickness of the coccolith shows a significant correlation with $\Omega_{Ca}$ at the
sample depth (r = 0.41), and with the concentrations of nutrients nitrate and phosphate at 50 m
(r = 0.44 and 0.4, respectively). Surprisingly, the mean length showed no significant correlation
to any environmental variables except with PAR (r = 0.35).
The results of RDA can provide another critical perspective on the control of
environmental variables on coccolith morphology. The RDA1 and RDA2 explain together 58.3 %
of the total variations in coccolith morphological data. The surface sediment samples, color-



coded by different depth intervals, are distributed along the axis of RDA1, which is the most
important and explains 54.6 % of the total variance (Fig. 5A). Among the environmental
variables, $\Omega_{Ca}$ shows the highest correlation to RDA1 (r = -0.67, p < 0.05). The results of both the
correlation analyses and the RDA show that $\Omega_{Ca}$ in bottom water is the most important
environmental variable driving the morphological dataset, which shows a high correlation with
mean ks (r= 0.69; p < 0.05) and could explain up to 47 % ($R^2$= 0.47) of the variance observed in
mean ks (Fig. 5B). The RDA2 explained 3.69 % of the variance and is mainly correlated to the
salinity, temperature, pH, phosphate, TALK, and $pCO_2$ (Fig. 5A). The null response of coccolith
length to any environmental parameter is also observed in the RDA plot by its position near the
center of the ordination space, significantly contrasting with other morphological parameters
(Fig. 5A).

Table 2. Correlation matrix (p-value and Pearson correlation) between biological and
environmental variables. Bold values indicate significant correlations (with p < 0.05).

| Environmental/ Biological | p value | | | | | r | | | | |
|---|---|---|---|---|---|---|---|---|---|---|
| | Mean ks | Mean thickness | Mean length | Mean volume | Mean mass | Mean ks | Mean thickness | Mean length | Mean volume | Mean mass |
| Salinity | 0.79 | 0.07 | 0.87 | 0.86 | 0.86 | -0.05 | 0.34 | 0.03 | -0.03 | -0.03 |
| Temperature | 0.94 | 0,04 | 1.0 | 0.90 | 0.90 | -0.01 | -0.39 | 0.00 | -0.02 | -0.02 |
| Phosphate | 0.88 | 0.03 | 0.63 | 0.96 | 0.96 | 0.03 | **0.41** | -0.09 | -0.01 | -0.01 |
| Nitrate | 0.36 | 0.02 | 0.30 | 0.71 | 0.71 | 0.17 | **0.44** | -0.20 | 0.07 | 0.07 |
| TALK | 0.53 | 0.13 | 0.70 | 0.60 | 0.60 | -0.12 | 0.28 | 0.07 | -0.10 | -0.10 |
| Chlorophyll-a | 0.26 | 0.18 | 0.88 | 0.18 | 0.18 | 0.22 | 0.25 | -0.02 | 0.26 | 0.26 |
| PAR | 0.28 | 0.05 | 0.06 | 0.50 | 0.50 | -0.21 | -0.38 | 0.35 | -0.13 | -0.13 |
| pH | 0.18 | 0.31 | 0.38 | 0.24 | 0.24 | -0.26 | 0.19 | 0.17 | -0.22 | -0.22 |
| $pCO_2$ | 0.16 | 0.33 | 0.38 | 0.21 | 0.21 | 0.27 | -0.19 | -0.17 | 0.24 | 0.24 |
| $\Omega_{Ca}$ | **0.00** | **0.03** | 0.05 | **0.00** | **0.00** | **0.67** | **0.41** | -0.36 | **0.66** | **0.66** |





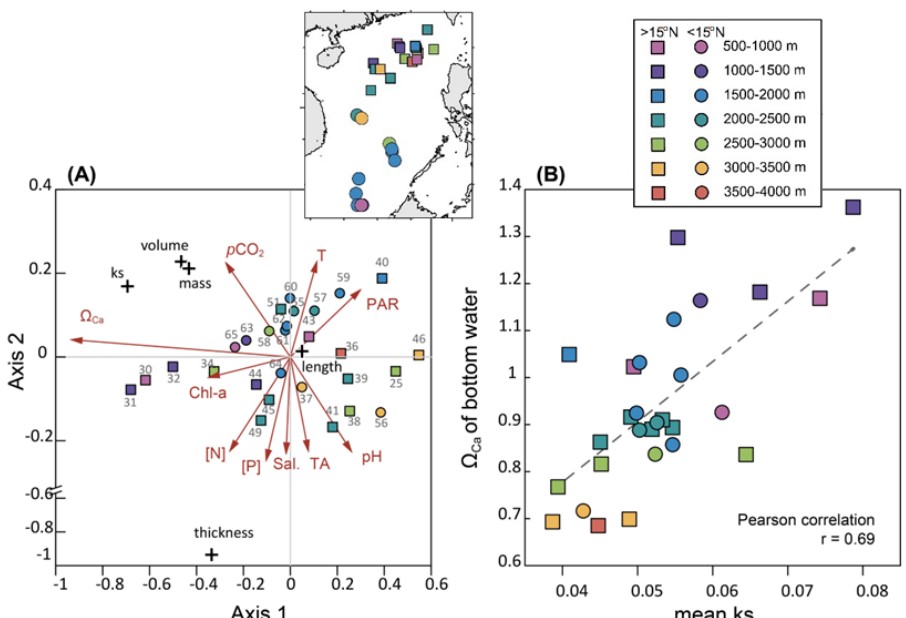


Figure 5. Redundancy analysis (RDA) ordinations for environmental variables and morphological

measurements (A) and (B) linear correlation and correlation coefficient ($p < 0.05$) between $\Omega_{Ca}$

at bottom depths and mean ks from surface samples.


## 5. Discussion

### 5.1 Comparations between laboratory dissolution experiment and natural samples


In this study, we evaluate fossil coccolith responses to dissolution under laboratory experiments and field settings. In the dissolution experiment, ks values are higher than the modern coccoliths in the SCS due to the higher abundance of the relatively thicker *G. caribbeanica* in the downcore sediment sample. Though the absolute values of ks cannot be directly compared between the dissolution experiment (Fig. 2B) and surface sediments (Fig. 4A), the trajectory of morphological variations during the dissolution experiment does provide important diagnostic information to explain phenomena observed in the surface sediment samples.

First, the phenomenon that coccolith length increased with the decrease of ks could be observed in both the dissolution experiment (Fig. 2C) and natural surface sediments (Fig. 4B).





The laboratory experiment showed that under controlled conditions (known changes in water
chemistry and uniform species composition), the coccolith morphology variations (mean length
and mean ks) reflected different degrees of dissolution. We also observed a length-related
dissolution pattern, where smaller coccoliths gradually dissolve with the increase in Calgon®
concentration, leading to a higher average length but a lower mean ks. The mean ks and mean
length relationships in the surface samples (Fig. 4B) show a similar trend to the laboratory
observations (Fig. 2C). Thus, the observed trend and the largest size and lowest ks in the surface
sediment samples are explained by the dissolution of the smallest species due to the lower $\Omega_{Ca}$
at the deepest samples and increasing the abundance of the larger coccoliths.

Second, changes in the σ/ks ratio in the dissolution experiment a slight and gradual

increase in dissolution and then a decrease with the highest concentrations of Calgon® (Fig. 2C).
In the laboratory experiment, the subsample with no Calgon® solution presented well-preserved
coccoliths with high mean ks and a small standard deviation. As the amount of Calgon® solution
added to each subsample increases, small coccoliths start dissolving preferentially, decreasing
the mean ks and increasing the standard deviation (Fig. 2B). With higher amounts of Calgon®
solution (4 and 6 ml), the small coccoliths are completely dissolved, resulting in an assemblage
dominated by larger coccoliths (Fig. 2C). Under these highest dissolution stages, the larger
coccoliths are also partially dissolved then both mean ks and σ decrease (Fig. 2B). In this way,
the σ reflects how differential dissolution size selection affects the composition of the
assemblages. Hence, samples that are more (less) susceptible to dissolution result in more
homogeneous (heterogeneous) assemblages regarding carbonate preservation.

However, the trajectory of σ/ks vs. ks in surface sediments seems more complex than in

the dissolution experiment (Fig. 4A). First, there is a group of samples with a positive correlation
between σ/ks and ks from shallow areas of the north and south SCS. The depth of samples from
northern and southern SCS regions is similar, so we argue that this feature is not caused by
dissolution but due to the species differences in both parts of the SCS. The coccolithophores
have multi-stage blooms in the north SCS, with a peak of *G. oceanica* in late winter, when
coccolithophore fluxes are highest due to strong water column mixing and renewed nutrient
inventory, and another of *E. huxleyi* in early spring (Jin et al., 2019; Chen et al., 2007). In
contrast, *E. huxleyi* is the dominant species in the more oligotrophic south SCS (Fernando et al.,
2007b) due to its higher competitiveness in situations of lower nutrient concentration
(particularly nitrate) compared to *G. oceanica* (Eppley et al., 1969). So, even without any
influence from dissolution, the coccolith in the north SCS should feature a higher species
diversity and, thereby, a higher σ/ks compared with the coccolith in the south SCS. Hence, the
variety of the coccolith assemblages in the surface sediment samples results in different





trajectories in σ/ks vs. ks plotting. But the general trend of σ/ks vs. ks in surface sediment is still
following the trends observed in the dissolution experiment: (1) ks decreases with dissolution;
(2) σ/ks increases slightly when dissolution starts and (3) then it decreases with greater
dissolution.

5.2 Sedimentary record of coccolith morphology: life-cycle vs. dissolution factors
Previous studies have evaluated changes in the calcification of Noëlaerhabdaceae
coccoliths in glacial-interglacial cycles through analyses of the coccolith mass and attributed
morphological variations mainly to water column nutrient availability and carbonate chemistry
parameters, related to the coccolithophores life-cycle (e.g., Beaufort et al., 2011). Su et al.
(2020) found that the environmental dynamics of the surface photic zone controlled
Noëlaerhabdaceae coccoliths' calcification in northern SCS (MD05-2904). Similarly, higher
calcite production recorded by increased coccolith mass has been attributed to the increased
[$CO_3^{2-}$] in the surface water column in the South Indian Ocean and North Atlantic Ocean in
modern sediments (Beaufort et al., 2011). Dissolution effects were thought to be less likely
drivers of changes in the morphology of coccolith (Beaufort et al., 2011; Su et al., 2020), which
is a reasonable assumption for the coccoliths depositing in shallow sediments above the
lysocline. These interpretations are partially sustained by the findings of Beaufort et al. (2007),
who found no significant coccolith dissolution during the settling in sediment traps deployed
between 250 and 2500 m. The former study proposed that most of the dissolution occurs in the
euphotic zone and possibly in the guts of grazers, therefore, discarding the impact of bottom
water chemistry and/or post-burial processes on coccolithophore weight.
In our set of samples in the SCS, the RDA results show that mean thickness and length
significantly correlate to nitrate and phosphate at 50 m (Table 2). This observation agrees with
Jin et al. (2016), who found that biometric attributes of *E. huxleyi* correlated with nutrient
concentrations in the plankton samples in the ECS. Nutrient variables are important for
coccolithophore calcification (Raven and Crawfurd, 2012) and morphological parameters, at
least in species of the Noëlaerhabdaceae family (Båtvik et al., 1997; Pasche et al., 1998).
However, based on the extended evidence of our study, including carbonate chemistry at the
depth of the sediment samples in the SCS, we observe evidence that several of the
morphological parameters measured are not only influenced by primary biomineralization. Still,
abiogenic post- or syn-depositional processes override this signal in the sediment samples in this
region. The highest correlations between coccolith morphology, namely mean ks, volume, and
thickness, with the bottom water calcite saturation, $\Omega_{Ca}$, indicates that the calcium carbonate
preservation conditions could strongly override some of the morphological parameters in fossil



coccoliths (Table 2, Fig. 5A). We suggest that the mean ks of coccolith could be a potential proxy
for the carbonate dissolution in the bottom water, especially in sites near or below the
lyscocline.

Carbonate dissolution may also happen within the shallow sediment (Sulpis et al., 2021;

Subhas et al., 2022). Based on our current dataset and using only the morphological variations,
we cannot distinguish where the dissolution happens at the time of deposition in the sediment
water interface, or post-burial in the first cms of the seafloor sediment. For the deep ocean
depositions with lower sedimentary rates, such as the deepest parts of the SCS (Huang and
Wang, 2006), the exposure time of particles to bottom water should be longer than that along
the continental slope. Thus, we suggest that the major dissolution in the deep SCS happens on
the sediment-water boundary instead of within pore water. Interestingly, the ks of coccolith in
the surface sediment of the ECS are much lower, as low as 0.01 (Jin et al., 2019), than those in
our study, which is higher than 0.04. However, the ks of coccolith during the laboratory
dissolution experiment performed by Jin et al. (2019; Fig. 9A in that study) show the same range
as our measurements. The ECS samples are from the continental shelf with high sedimentary
rates and organic carbon content (Jin et al., 2019). In these settings, the coccoliths continuously
dissolve after being buried within the first centimeters of the seafloor sediments in response to
organic matter remineralization and $CO_2$ release, resulting in a ~30-50 % decrease in coccolith
mass (Jin et al., 2019). Therefore, the sedimentary environment has to be individually evaluated
to understand which process is controlling the dissolution of coccolithophores at the seafloor.
More detailed work, such as in-situ pore water chemistry measurements, would be necessary
to fully reveal the fate of coccolith dissolutions in different burial scenarios (Holcová and
Scheiner, 2022).

Among all the morphological parameters, we find the mean ks of coccolith as a more

robust dissolution proxy compared to the other measured morphological parameters. Firstly,
we observe a higher correlation coefficient between mean ks of coccolith and $\Omega_{Ca}$ compared
with other morphological parameters. Secondly, although volume, mass, and thickness are also
highly correlated with $\Omega_{Ca}$, these morphological parameters vary more with the feature of
different coccolithophores, including variations in coccolith circularity and cell sizes (Young and
Ziveri, 2000; Bolton et al., 2016). Thirdly, the thickness is a morphological pattern sensitive to
the upper ocean's preservation and surface ocean environmental conditions during
biomineralization (Table 2). Another important feature of ks is its high sensitivity to dissolution.
As shown in Fig. 4, the ks of coccolith have already begun to decrease even though the water
depths are only at ~2000 m, which is below the modern lysocline but above the CDD in the SCS
(Wang et al., 1995; Luo et al., 2018). Finally, the dissolution effects on morphological attributes



of mean ks agree well with the laboratory dissolution experiment, in which each subsample's
mean ks reflected different preservation stages (Fig. 3).
Despite a noticeable degree of uncertainty due to the mixing of life-cycle and post-
mortem signals in the sedimentary record, similar findings of calcite dissolution modifying
coccolith's morphology in waters at or below saturation suggest that the conclusions drawn
from the present study are not unique to the SCS. In the Sub-Antarctic and Antarctic zone,
dissolution signals affecting coccolithophores were manifested as a decrease in mass and distal
shield length of *E. huxleyi* coccoliths preserved in surface sediments (Vollmar et al., 2022; Rigual-
Hernández et al., 2020b). Based on this collective evidence, a key reasonable question could be,
"can the morphological variation of coccolith be employed to trace the evolution safely, or could
they be a good proxy for carbonate preservation"?

5.3. Implications for interpreting the downcore history of coccolithophore morphology
On longer time scales, the morphological variations of coccoliths have been employed
to trace coccolithophores evolutionary trend. Bolton et al. (2016) first measured the ks of
Noëlaerhabdaceae in the last 15 million years. They found that the decrease of coccolith ks
paralleled the reduction of atmospheric $pCO_2$ since the late Miocene and interpreted this as a
decrease in biomineralization. More recent works by Beaufort et al. (2022) and Jin et al. (2022b)
focused on the coccolithophore evolution over the last 2 million years by measuring coccolith
mass, highlighting the role of seasonality and local environments in the evolution and
production of Noëlaerhabdaceae. Similarly, Guitián et al. (2020) studied size trends across
different regions between Oligocene to the Early Miocene, concluding that cell size distribution
was controlled by multiple competing factors, with a strong selective pressure from $CO_2$ decline
a potential mechanism. This study examined dissolution by looking, among others, at the
fragmentation and etching of coccoliths and found that temporal trends in lith size distributions
were not significantly affected. This agrees with our observations since the mean length in SCS
surface sediments does not correlate with any saturation state related parameter. However,
Guitián et al. (2022), using a new calibration in the C-Calcita software that enables estimations
of coccolith thickness up to 3.1 microns, found that elliptical ks (kse) was inversely correlated
with the relative abundance of dissolution-resistant nannoliths. This was interpreted as a
dissolution control on the elliptical shape factors in coccolithophores between Oligocene and
Miocene, as it was found in our surface sediment samples. Therefore, we propose that for
studies focusing on coccolithophores evolutionary histories, it would be safer to select a shallow
sediment core with low organic carbon content, high clay content, and always lying above the
carbonate lysocline (Guitián et al., 2020).



One useful way to identify dissolution in these studies covering geological time scales

could be plotting the $\sigma/ks$ against ks. If an increase of $\sigma/ks$ is detected in the sediment coccolith
without any significant variations in coccolith assemblage or with an increase of dissolution-
resistant species (Guitián et al., 2022), it should be interpreted as a dissolution. Another way to
determine separate evolutionary/ecological influences on ks variations could be to measure the
ks of coccolith across a close spatial gradient which includes different depositional depths.
Significant variations in the morphological attributes of the fossil coccolithophores would likely
be caused by different saturation through time at different sites. Related to this last suggestion,
coupling downcore morphological assessment in coccolithophores with other calcareous
proxies measurements, such as size-normalized weight of planktic foraminiferal tests (Lohman,
1995; Broecker and Clark, 2001; Barker et al., 2002), which include recent advances in
morphological analyses in large microfossils (Iwasaki et al., 2015; 2019), may provide an even
more precise and safe quantitative estimates of past deep-carbonate chemistry parameters.

## 504  6. Summary

This study demonstrates, based on morphological attributes of *E. huxleyi* and
*Gephyrocapsa* spp. (> 2 μm), that dissolution effects primarily affect the morphology of
coccoliths preserved in the deep ocean. In the SCS surface sediments, bottom water $\Omega_{ca}$
saturation plays a major role in the variation of the coccoliths' ks shape factor, which has the
potential, based on the current calibration, to quantitatively reconstruct past carbonate
dissolution changes. Our laboratory-controlled dissolution results show that the mean ks
decreased in response to increased amounts of corrosive solution. We propose the ratio $\sigma/ks$
vs. mean ks to evaluate the degree of dissolution (light, strong, or no dissolution) occurring in
the sedimentary record. A length-related dissolution pattern was also observed in the laboratory
and surface sediments, with small coccoliths more prone to suffer dissolution, increasing larger
coccolith specimens and affecting the assemblage composition. As in the laboratory experiment,
the coccolith's ks from surface sediments decreased with dissolution, and the $\sigma/ks$ trajectory
reflected different dissolution stages. However, the $\sigma/ks$ in surface sediment showed a more
complex response due to the natural variability of the surface sediment samples in terms of
geographical differences in multiple environmental factors. These findings demonstrate that,
despite the complexity of the carbonate system and ecological aspects, the coccoliths ks factor
allied to $\sigma/ks$ ratio has potential as a dissolution proxy to track changes in the seafloor carbonate
saturation state. Although a stable proxy, the mean ks should be applied with caution,
particularly when applied to longer time scales, in which evolutionary trends might exert control
on morphological attributes of fossil coccolithophores.



## Author contributions

AG, HZ, RHN and IHA conceived and designed the study. AG and HZ conducted the lab work and sample analyses. AG, HZ and IHA performed the statistical analysis. AG, HZ and IHA wrote the manuscript with substantial contributions from all co-authors.

## Competing interests

The authors declare that they have no conflict of interest.

## Data availability

Research data is available as supplementary material and in the Zenodo (https://doi.org/10.5281/zenodo.7271441, Gerotto et al., 2022) and PANGAEA (pending doi) data repositories.

## Acknowledgments

This study was financed in part by the Coordenação de Aperfeiçoamento de Pessoal de Nível Superior - Brasil (CAPES) - Finance Code 001 to A.G. and the ETH Core and Swiss National Science Foundation (Award 200021_182070) funding to H. M. Stoll. Additional funding was provided by the National Natural Science Foundation of China (grants 42188102 and 41930536) to C.L. Thanks also to the International Ocean Discovery program for providing the sample used for the dissolution experiment.

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
