# Peer review of "Responses of fossil coccolith morphology to preservation conditions in the deep ocean"

_EGUsphere, 2022_

## Author Response (AR1)

Dear Editor,

We would like to thank the reviewers and editor for the valuable comments and suggestions. We have taken them into consideration and made modifications in the manuscript to address them. In order to address the comments we made changes throughout the manuscript and in the Figures, which are highlighted point by point in the list below, considering specific comments of the editor followed by the reviewers' comments. The lines listed in the authors' response refer to the revised manuscript.

Sincerely yours,

Amanda Gerotto, PhD.

Center for Marine Studies
Federal University of Paraná
Av. Beira Mar, s/n
Pontal do Paraná, PR,
Brazil 83255-976
E-mail address: gerottoamanda@alumni.usp.br

**Associate editor:**
Three reviewers provided comments to the manuscript. All the reviewers agreed that the study is interesting, the results are supported by a well written M&M section, and the discussion is well-structured and supported by the data presented in the text. Every reviewer provided some general suggestions for improvement, as well as some technical corrections. Collectively, the reviewers suggested to change the title to better emphasize the development of a new proxy and recommended to:

**1) elaborate more on the possible species/assemblage influence on the global records of coccoliths;**

**Authors response:** This topic is further discussed in response 7, below.

**2) possibly re-perform the RDA analysis to avoid including redundant variables;**

**Authors response:** We now specify in line 312 that the autocorrelated variables were kept considering their strong influence on coccolith morphology during the life-cycle. Although redundant, these variables are related only to surface ocean parameters, which would inflate the variance explained by  surface ocean variables. Nevertheless, our results showed that the deep ocean variable is the most significant, so the redundancy of surface ocean variables does not have an impact on the final results.

**3) test the proxy against an independent dataset; and**

**Authors response:** This topic is further discussed in response 7, below.

**4) better highlight the role of dissolution when the ks factor and/or thickness are used for evolutive studies.**

**Authors response:** As we stated in our response to RC3 we carefully emphasized in the Section 5.3 the role of dissolution for studies focusing on evolutions of coccoliths, that it's better to

check the preservation of coccolith before treating ks as a result of evolution. That's also one of the main conclusions of our work.

**I think that all these recommendations are worth being considered, which the authors did based on their reply to the reviews received.**
**In addition to the comments by the reviewers, I have some additional suggestions that I would like to propose to the authors.**

**1) I believe that it would be helpful if the sample locations were specified in panels B-I of Figure 1.**

**Authors response:** The sample locations (depth and latitude) are now plotted in panel B. We chose to keep the symbols only in one panel so as not to impair the visualization of the profiles by adding too many data points.

**2) Section 3.3, please refer to Figure 1, as this figure shows many of the parameters mentioned in this section.**

**Authors response:** Modified. Panels B-I of Figure 1 are now mentioned in section 3.3.

**3) Figure 3 – is there any correlation between the results obtained and the number of measurements conducted on each sample? The authors should comment on this in the main text.**

**Authors response:** That is a really nice question. The coccolith ks are very scattered for a surface sediment sample. That means the sampling number inevitably influence the final results. This point has been largely ignored in previous coccolith morphology studies. Some works only measured 50 specimens, while other works could measure up to thousands for one sample. And all studies claimed that their number of measurements is safe. In a recent published paper by Zhang et al., 2023 Marine geology, more than ~1500 coccoliths were measured per sample. Then, authors resampled the measurements and recalculated the mean ks. As shown in the Figure below (Zhang pers. comm.), the mean ks gradually converges to the "real" mean ks (black dashed line) with the increase of number. In our study, we measured 100-400 coccoliths, with an average of 250 measurement per sample. The samples with 100 measurements may have a larger error. But these samples also had a lower number of coccoliths as they were affected by stronger dissolution. Considering that according to Zhang et al (2023) there is a ~10% of variability in the measured ks with ~100 measurements, and it goes to a minimum of ~5% of variability if the measurements go up to 600 measurements, we argue that at least 100 measurements is a safe minimum. In addition, we suggest that 250 measurements per average in our study provides a good compromise between time and robustness of our results, which result in an uncertainty of ±~0.004 in the ks. Considering the ks in our samples ranges from 0.04 to 0.08, this uncertainty does not compromise our results. We plan to continue performing additional tests on this matter in order to provide a more quantitative estimate of uncertainties using different techniques to quantify past coccolithophore morphology.

[Figure]

Figure showing the number of measurements and variation in the mean ks performed for samples from IODP Site U1433, in the South China Sea (Zhang pers. comm).

**4) Table 2 – in the caption, please provide a definition for TALK, PAR, and $\Omega$Ca.**

**Authors response:** The definitions are now included in the caption.

**5) About the results of the dissolution experiment – from the description of the results, it appears to me that the authors conducted the experiment at room temperature. If they were to conduct the experiment at lower temperature (2-4 C, for example), which are more realistic for deep ocean settings, would they expect to see a difference in the experiment outcome? Please add a comment on this in the main text.**

**Authors response**: Thanks for this suggestion. We have added the temperature information in this new version and discuss temperature effect in this new version.
The omega-calcite is the only real important parameter in all dissolution experiments. A dissolution of calcite can be achieved by decreasing omega-calcite via adding acid (removing $CO_3^{2-}$), decreasing temperature, increasing pressure or removing $Ca^{2+}$. In this work, we dissolved the carbonate by removing $Ca^{2+}$, and this method is no difference with other ways to trigger a dissolution. So, in this case, simulating the deep ocean environment conditions is not very necessary, but also difficult to accomplish. We need not only decrease the temperature to ~2˚C, but also increase the pressure to about 100-350 bar (which requires special equipment not common in a conventional micropaleontological lab). If we decrease the temperature from 25 to 2 ˚C, more dissolution would occur due to the increase of gas solubility in water and higher $CO_2$ concentration (increasing acidification), which would have similar effects as the removal of $Ca^{2+}$. Instead of explaining the dissolution results directly, we add one sentence for the deep ocean sediment in section 4.2 (line 290):
*"In general, the degree of dissolution varied according to the depth of the sediment samples. The calcite saturation, $\Omega_{Ca}$, decreases with colder temperature, higher pressure and higher $CO_2$ concentration in deep ocean."*

**6) End of section 5.2. I think that it would be beneficial if the authors were to expand more on the comparison with other studies in other geographic locations. In their reply to Reviewer #2,**

**the authors stated that they cannot evaluate how well their proxy would predict bottom omega calcite using an independent dataset. Because of this, I think that a more comprehensive comparison of their results with results obtained from other geographic locations will help the authors emphasizing the validity of their newly developed proxy.**

**Authors response:** Yes, we agree that it would be great to compare with other works. However, our work is the first and only careful comparison between coccoliths morphological parameters and deep ocean carbonate chemistry. The only study measuring coccolith morphology in the East China Sea (Jin et al., 2019 MM) did not report the carbonate chemistry. And the carbonate chemistry data on continental shelf are not available from other global datasets, such as GLODAP.

In our response to Reviewer 2, we were conservative, not pessimistic, to our results. We therefore prefer to focus on the results of this single basin, with unique biological and biogeochemical characteristics. Further studies will be necessary to corroborate our findings in other regions and to produce an universal calibration of ks as a proxy of $\Omega_{Ca}$. In the last 15 years, the shape of coccoliths has been assumed to be mainly controlled by surface ocean carbonate chemistry parameters. And in the last two years, new studies are now reporting that this relationship between morphological parameters (ks in particular) and surface ocean carbon chemistry is not unequivocal and most likely regional dependent (Jin et al., 2020, Vollmar et al., 2022, Guitián et al., 2022). So, in this work, we want to deliver the following message to the scientific community: dissolution is really important for coccoliths recovered from the deep ocean. We are looking forward that our work can stimulate more research groups generating comparable data to better calibrate this proxy globally in the future.

**7) Finally, I think the paper would benefit from a SEM plate where the authors show the degrees of dissolution as discussed in the text. In the plate, the authors might even add a little drawing where they to summarize the measured parameters.**

**Authors response**: We think summarize the dissolution feature could be a very good idea, if we use the completeness of coccoliths (the percentage of broken coccolith) as a dissolution proxy. Unfortunately, SEM pictures are not available because all samples have been consumed already. This has been described in Line 170. Moreover, a SEM has advantages in identifying broken coccoliths or malformations of coccoliths. At the early stage, the second author (Dr. Hongrui Zhang) tried to use the completeness of coccoliths under SEM as a proxy for dissolution (see the figure below how we try to classify different coccoliths). However, this is very hard approach and subjective. This quantifying problem has been perfectly solved by the estimation of coccolith thickness (or mass or the ks), which is more objective. So, we focus on the **thinning of coccoliths**, instead of **breaking**, in this study. And this measured parameter cannot be clearly illustrated in a SEM picture. In recent years, light microscopes with well-calibrated light source work better than SEMs in quantifying coccoliths' dissolution. We suggest that a light microscope in combination of automated image techniques such as the one used in this study should be the priority for that specific work.

[Figure]

**Overall, I invite the authors to resubmit a revised version of their manuscript after what I view to be a moderate revision.**

**References:**

Guitián, J., Fuertes, M. Á., Flores, J. A., Hernández-Almeida, I., and Stoll, H. Variation in calcification of Reticulofenestra coccoliths over the Oligocene-Early Miocene, Biogeosci. Discuss., 1-17, https://doi.org/10.5194/bg-2022-66, 2022.

Vollmar, N. M., Baumann, K. H., Saavedra-Pellitero, M., and Hernández-Almeida, I. Distribution of coccoliths in surface sediments across the Drake Passage and calcification of *Emiliania huxleyi* morphotypes, Biogeosciences, 19(3), 585–612, https://doi.org/10.5194/bg-19-585-2022, 2022.

Zhang, H., Zhou, X., Jiang, X., Hernández-Almeida, I., & Liu, C. (2023). The source of Pleistocene carbonate below the CCD in the central basin of South China Sea: Evidences from coccolith and geochemistry. Marine Geology, 107011.

**Reviewers' comments:**

**Authors response (RC1):**
We would like to thank the reviewers and editor for the valuable comments and suggestions. We have taken into consideration both questions and specific review concerns. The questions were answered below each reviewer's comment. Considering RC1 specific comments we followed all the reviewer's advice. To address them changes were made throughout the text.

**RC1**
**The project is well designed and has produced some interesting results along with a well-written manuscript to accompany it. The project has identified a novel proxy for quantifying rates of carbonate dissolution with valid methods clearly outlined. Results are clearly displayed in figures and tables with strong explanations of the proxy application as well as cautions.**

**Questions for authors to address if they feel it would add to the story:**
**It would be helpful to hear whether these were the results expected by the authors.**

**Authors response:** The expected dissolution pattern results (mean ks) were described in lines 219-223. However, to complement this topic, additional remarks regarding the expected results regarding dissolution primarily affecting the morphology of coccoliths were added in line 396. The relationship between bottom water carbonate chemistry/dissolution and coccolith morphology is as what we expected. Before carrying out the RDA, we thought the surface

processes' impacts could be larger, but the results indicated that coccolithophore growth in the surface ocean only plays a limited role in coccolith thickness on a basin scale.

**The abstract mentions that degree of dissolution and size-selective dissolution is influenced by assemblage composition but this is not fully addressed in the text.**

**Authors response:** We complement the paragraph between lines 396 and 398 highlighting the role of assemblage composition on the degree of dissolution according to the large geographical variability influencing the coccolithophore calcite production during its life cycle.

**Could elaborate on species/assemblage influence – fragility due to size, crystal composition etc. this is left rather vague.**

**Authors response:** We did not carry out any dissolution experiments on the species' influence. We only evaluated the role of the assemblage composition, as testing the effect of crystal composition with requires additional analyses and instrumentation that was beyond the original goal of this study. But we believe that the species/assemblage difference could be mainly caused by the fragility difference between *G. oceanica* and *E. huxleyi*. We made this hypothesis clear in lines 175-177, 309, and 352. The downcore assemblage differs from the surface sediment samples in their higher proportion of the thicker coccolithophore species *G. caribbeannica*, compared to the thinner *G. oceanica* and *E. huxleyi*.

**How might this measurement influence global records of coccoliths?**

**Authors response:** Mean ks combined with σ/ks vs. ks can be applied to global records since this new index considers the different compositions of assemblages according to geographical variability. We suggest the principle rule described in our work should be universal in other basins. However, caution still should be kept in mind. For example, the assemblage measured in this work was mainly composed by *E. huxleyi* and *G. oceanica.* How the ks and σ/ks behave in coccolithophore assemblages characterized composed by different species (today or in the past) should be tested. Moreover, should we use a mono-species morphology parameter, or we can mix all coccoliths even from different family? These should be done in the future works in the next few years.

**Specific comments by line:**

**Authors response:** All the following suggestions have been accepted, or replied to if a longer explanation was needed.

11 – critical to elucidating
14 – complex not complexity
15 – during an organism's life cycle
21 – samples from the South China Sea
22 – surface sediments were
24 – statistical analysis indicates that
39 – ocean $CO_2$ is influenced (atmospheric $CO_2$ = $pCO_2$)
42-43 – concentration, and carbonate
56 – variations in the ocean carbon
65-66 – provides a quantitative
73 – called coccoliths. Coccoliths
74 – up to 80 % of deep-sea
75 – changes in coccolith morphology are believed

89 – there has been no study
96 – between coccolithophore biometry
97 – building on these results
99 – it has also been demonstrated
104 – studies that systematically explore the drivers
121 – by shallow passages to the north and south
122 – water exchange between
125 – East Asian Monsoon (EAM; Wang and Li, 20009)
148-149 – relatively low DIC and TALK and high pH
169 – add reference for smear slide preparation technique
170 – dissolution experiments using
171 – obtained from a Late Pleistocene
173 – what is the thinner species that is being referred to?
**Authors response:** We added in the text "compared to the thinner species (e.g. *E. huxleyi*)"
175 – suspension was separated into
176 – each with a volume
177 – has traditionally been used
198 – parameters of coccoliths in the
209 – calculated using the formula by Young
210 – obtained from C-Calcita
229 – coccolithophorid is observed in
306-307 – between several coccolith morphological parameters and bottom
360-361 – rephrase
**Authors response:** We fixed a typo in the sentence. Now it reads "Second, changes in the σ/ks ratio in the dissolution experiment reflect a slight and gradual increase in dissolution and then a decrease with the highest concentrations of Calgon® (Fig. 2C)."
427-428 – deep ocean deposits with lower sedimentary
442 – coccolith dissolution in different
444 – ks of coccolith is a more
465 – variation of coccoliths be employed
470 – to trace evolutionary trends
488 – focusing on coccolithophore evolutionary histories
493 – increase in dissolution
494 – interpreted as dissolution
514 – more prone to dissolution (without "suffer")

**Continuity:**

**Vs or vs.? Should it not be *versus/vs* ?**

**Authors response:** We chose to use *vs.* and changed it throughout the text.

**Sea floor or sea-floor?**

**Authors response:** We chose seafloor and changed it throughout the text.

**Authors response (RC2):**

We want to thank the reviewer for their valuable comments and suggestions. We have considered them and made modifications to the manuscript to improve it. To address the reviewers' specific comments changes were made throughout the text. Specific questions were answered detailed below the reviewer's comment.

**RC2**

**General comments:**

In this manuscript, Gerotto et al. make use of dissolution lab experiments and sediment samples for develop a proxy for the reconstruction of past carbonate dissolution dynamics. For do that, they compare morphological measurements of coccoliths came either from modern surface sediments along basin-scale environmental vertical gradients as those resulting from dissolution experiments using sediment samples taken elsewhere on the Pacific. The thematic thread conveys the reader naturally to the theme under study. The Theoretical background is comprehensive but concisely enough to give support to the discussion. The methods are described in-depth and are suitable for addressing the aim of the study. The Results are properly weighted into a well-structured Discussion. They properly recognize in M&M and Discussion that the sensitivity resulted from dissolution experiments and modern samples cannot be compared directly, as well as, has critically described the effects of Calgon® solution in carbonate particles. Therefore, after minor reviews posted below are addressed, I find this manuscript is suitable for publication in Egusphere.

**Specific comments:**

**Title – Much more straightforward if it includes that a new proxy was developed**

**Authors response:** To address this comment, we have modified the manuscript title to "Fossil coccolith morphological attributes as a new proxy for deep ocean carbonate chemistry".

**Figure 1 –Include a larger inset map; In captions remove source of the data and direct the reader to M&M**

**Authors response:** We have modified the figure and caption.

**- In the RDA model it's appear to be redundant variables (ex. the TA-Sal, pH-pCO2 and N-P pairs of variables are expected to be strongly autocorrelated as Fig. 5a actually shows) that might be introducing statistical noise and eventually reducing % of explained variance and/or impeding a more direct evaluation of mayor environmental drivers on coccolith morphology. If you think it could be the case, apply a test for identify redundant variables (ex. varclus procedure in RStudio) and redo the RDA analysis including only non-redundant variables.**

**Authors response:** We performed a correlation test at the beginning of the statistical analysis between temperature, salinity, phosphate, nitrate, silicate, alkalinity, dissolved inorganic carbon ($TCO_2$), pH, fugacity of CO2 ($FCO_2$), partial pressure of $CO_2$ ($pCO_2$), $HCO_3$, $CO_3$, $CO_2$, Total Boron, OH, revelle factor, chlorophyll-a concentration, photosynthetic active radiation, and omega calcite in the bottom. We removed some of these variables due to autocorrelation. We chose to keep some autocorrelated variables as they strongly influence coccolith morphology during the life-cycle (Chen et al., 2007; Jin et al., 2016).

**- Mention in the discussion the environmental data used was not obtained in-situ but from climatologies including interpolations, etc.**

**Authors response:** We carefully described the feature of data in the method section. We now specify at the end of line 230 and referring to the environmental data "were extracted from different databases, interpolated to the geographical location of the surface sediment samples."

**- It's possible to evaluate how well your proxy predict bottom omega calcite using an independent dataset?**

**Authors response:** Unfortunately there are not independent datasets using the same morphological parameters in coccoliths (ks) with the same method (circular polarization and C-Calcita) in modern samples which could be used to validate our proxy. We should keep in mind that this proxy would not work in samples located along small gradients of deep water carbon chemistry, well above the lysocline, as it is mentioned in section 5.2. In addition, there are other potential drivers of coccolith dissolution (and variation of morphological parameters) such as changes in the DIC as the result of organic matter respiration at the seafloor, so the application of this parameter to another dataset can not be done without considering these other factors.

**107 – Remove (n = 28) from the Introduction**

**Authors response:** Changed.

**360 – It's seemed a word as "caused" is missing**

**Authors response:** 'Caused' has been added.

**451 - Replace "environmental conditions" by "nutrients conditions"**

**Authors response:** Changed.

**465 – 466 elaborate better the question "…to trace their evolution safely, or instead be a good…"**

**Authors response:** We changed the text to ""can the morphological variation of coccoliths be employed to trace their evolution safely, or instead be a good proxy for carbonate preservation"?"

**476-479 – Elaborate better the end of this paragraph**

**Authors response:** we found there was a typo in the original manuscript and that is why it was not clear. Now the end of this paragraph reads as follows "with strong selective pressure from CO2 declines as a potential mechanism."

**520 – Maybe "complementarity" could be more precise than "complexity"**

**Authors response:** We changed to "complexity".

**Authors response (RC3):** We would like to thank the reviewer's valuable comments and suggestions. We have taken into consideration both specific comments and technical corrections. To address them changes were made throughout the text. Some specific comments were answered detailed below the reviewer's comment.

**RC3**
**In this manuscript, the authors deepen on the issue of how the dissolution affects the coccolith morphology and calcification by using both targeted lab experiments and sediment-core sample from a natural setting. The experimental approach, as well as the statistical treatment of the data is accurate and well developed. This work brings out interesting results and useful insights for reflection when it comes to evaluate the role of coccolithophore within the carbonate production, as well as to use the fossil assemblages in reconstructing past evolution and/or oceanographic conditions. The outcomes are well displayed and robustly discussed in**

**the manuscript. Although, I report below some specific comments to be addressed by the authors, and some technical corrections that need to be fixed in the text.**

**Some specific comments:**
**- The title should reflect better the novelty of this research, by adding a reference to the new index suggested here.**

**Authors response:** To address this comment, we have modified the manuscript title to "Fossil coccolith morphological attributes as a new proxy for deep ocean carbonate chemistry".

**- 487-490: To observe the evolutionary trends it is important to study long-time intervals (see Beaufort et al 2022à 2 Myr, Bolton et al 2016à 15 Myr). How long can be the time covered by the shallow sediment cores? I think it is better to state that it is necessary to pay attention to the bias that can be introduced by the dissolution when it comes to use the ks factor and/or thickness for evolutive studies.**

**Authors response:** The age of core top samples are less than two thousand year based on the report of Sonne cruises 95 and following publications (e.g. Wang et al., 1999). So, the feature of coccoliths in the surface sediment is mainly controlled by dissolution in deep ocean and ecology in surface ocean (minor role). What we have emphasized in the Section 5.3 is that, for studies focusing on evolutions of coccoliths, it's better to carefully check the preservation of coccolith before treating ks as a result of evolution. That's also one of the main conclusions of our work.

Ref.
Wang et al., 1999, Geophysical Research Letters, Holocene variations in Asian monsoon moisture: A bidecadal sediment record from the South China Sea,

**- Suggestion: maybe the authors could take into account to attribute a specific short name to the new dissolution index "ratio σ/ks vs. mean ks". The advantages would be: i) to characterize better the index and make it more "recognizable" among the community; ii) to make the text easier to read.**

**Authors response:** We attribute the "normalized ks index" short name to the dissolution index.

**Technical corrections**

**Authors response:** These technical corrections have been accepted unless it is specified.

- 14: complex
- 26: vs has to be written in italics
- 75: coccoliths morphology, distribution and abundances
- 144: "ODV" State the entire "Ocean Data View" when mentioning it for the first time

**- 172-174: specify the relative abundance of G. caribbeanica, what are the other "thinner" species anf their abundance.**

**Authors response:** The Noelaerhabdaceae family coccoliths in the sample ODP 807 is composed by 41% *G. oceanica* (>4μm), 34% *G. caribbeanica* (~3-4μm) and 23% Gephyrocapsa <3μm. We add the percentages to the text and specify the thinner species found in the SCS in line 173-177:

"The distribution of coccolithophore species belonging to the Noelaerhabdaceae family in the sample ODP 807 is 41% of *G. oceanica*, 34% of *G. caribbeanica* and 23% of small *Gephyrocapsa*. These taxa are thicker particularly *G. caribbeanica,* than  the thinner Noelaerhabdaceae species commonly found in the SCS (e.g. *E. huxleyi,* Roth and Berger, 1975; Roth and Coulbourn, 1982).

- 236: erase "extracted variables"

- 279: (e.g. 17930)

**- Figure 3: I would change the x axis with the depth, instead of using the site ID, which is more meaningful for the discussion of the data. In this way I would erase also the arrow pointing the increasing depth. Then, recall the table 1 in the caption.**

**Authors response:** We chose to apply color coding to the bars using the same pattern as in the following figures. In this way, the depth and the sample code can be identified easily in the figure. We also removed the arrow pointing to the increasing depth.

**- 271: I would change the title of this section linking this more to the results, as it is it is more related to a discussion section connecting the morphological data directly with the environmental factors. Change with something more like "Morphological changes in natural conditions"**

**Authors response:** We changed the 4.2 section title to "Variations in coccolith morphology in natural conditions".

**- 287: be consistent when using "versus" along the entire text. I suggest to always use *vs.***

**Authors response:** We chose to use *vs.* and changed it throughout the text.

**- 338: Change with "comparison"**

**Authors response:** Changed.

**- 376: species difference à probably meaning "assemblage composition"? Please be more specific.**

**Authors response:** We changed to assemblage composition.

- 383-384: change "coccolith" with assemblages

**Authors response:** Changed.

**- 391: I would not use the "life-cycle" in the section title as it is not discussed in depth, but just briefly mentioned. Please change the section title according to the main point presented in section 5.2.**

**Authors response:** We have modified the section title to "Sedimentary record of coccolith morphology: calcification *vs.* dissolution factors"

**- 411 and 435: ECS à state the acronym when mentioned for the first time, but I guess that the authors meant SCS.**

**Authors response**: The acronym refers to the East China Sea. The full mention was included in line 424 when first mentioned.

**Additional comments by the authors**

We added the doi that were pending from the Pangaea repository (line 548). We also fixed the manuscript carefully for typos.

---

## Referee Report (RR1)

In this version 2, the authors have settled the referees' observations that improve the manuscript, although to be published I recommend to produce a new version incorporating minor corrections that are listed below.

Line 18: Given "Gephyrocapsa spp., > 2 µm" actually comprise all the known Gephyrocapsa species coccolith sizes, I recommend to replace it throughout the text by "small Gephyrocapsa species" or similar.

Line 20: Delete comma after (ks)

Line 29: Relax the "all variations" statement using "major" or "main"

Evaluate migrate the section 2 into M&M

Line 235: After concentration add "used as a proxy of phytoplankton"

Line 239: Replace Ph by pH

Figure 2: Add tick marks at the x-axis

Figure 3: Homogenize the minor tick marks

Figure 4: Delete the minor tick marks

Line 312: Indicate what parameters are autocorrelated

Figure 5: Fix the values label to tick marks, as well as, include in parenthesis the % variability explained by each RDA axes

Line 392: A more recent study that Eppley et al., 1969 comparing the two species growth at different nutrient levels is: New Zealand Journal of Marine and Freshwater Research, 1995, Vol. 29: 345-357. Coccolithophores Gephyrocapsa oceanica and Emiliania huxleyi (Prymnesiophyceae = Haptophyceae) in New Zealand's coastal waters: characteristics of blooms and growth in laboratory culture.

Lines 396-398: Elaborate better this sentence as sounds recursive or meaningless

Line 408: The life cycle is not mention in Beaufort et al. 2011 study, but with species-environmental responses most related to shift in assemblage composition

Line 470: The statement seems correspond to Figure 4

Lines 478-479: Delete quotation marks

---

## Author Response (AR2)

Dear Editor,

We would like to thank the reviewers and editor for the careful revisions. We have taken all the comments and suggestions into consideration. We made modifications in the manuscript and figures to address them. All the changes are described point by point in the list below. The lines listed in the authors' response refer to the revised manuscript.

Sincerely yours,
Amanda Gerotto, PhD.

Associate editor decision: Publish subject to minor revisions (review by editor)
by Chiara Borrelli

Public justification (visible to the public if the article is accepted and published): The manuscript went through another round of reviews. We have now received three reports. All the reviewers agree that the manuscript is suitable for publication in BG after some suggestions are addressed. I invite the authors to incorporate the technical corrections provided by the reviewers in a revised version of their manuscript. During their revision, they should also address the suggestions provided by Reviewer #2.

**Report 1**

This manuscript by Gerotto et al. has made necessary changes including the title which now highlights the novel proxy developed and changes are also notable in the figures with the continuity of color clearly displaying depth throughout the paper. The manuscript is well-written and the research clearly explained in the text. A few minor changes should be made but I feel this manuscript is suitable for publication.

**Report 2**

In this version 2, the authors have settled the referees' observations, although to be published I recommend to produce a new version incorporating minor corrections listed below:

Line 18: Given "Gephyrocapsa spp., > 2 μm" actually comprise all the known Gephyrocapsa species coccolith sizes, I recommend to replace it throughout the text by "small Gephyrocapsa species" or similar.

**Authors response:** We changed to "*Emiliania huxleyi* > 2 μm, and small *Gephyrocapsa* spp." in line 28 and 206.

Line 20: Delete comma after (ks)

**Authors response:** Deleted.

Line 29: Relax the "all variations" statement using "major" or "main"

**Authors response:** We changed to the "the main variations".

Evaluate migrate the section 2 into M&M

**Authors response:** We evaluated and decided to keep the "2. Oceanographic settings" section as it is, preceding M&M.

Line 235: After concentration add "used as a proxy of phytoplankton"

**Authors response:** We added "(used as a proxy of phytoplankton)"

Line 239: Replace Ph by pH

**Authors response:** Replaced.

Figure 2: Add tick marks at the x-axis

**Authors response:** We included the tick marks at the x-axis.

Figure 3: Homogenize the minor tick marks

**Authors response**: We changed the range of minor units on the mean length, volume, and mass charts.

Figure 4: Delete the minor tick marks

**Authors response:** The minor tick marks were removed. We have also standardized the font used for the axis titles.

Line 312: Indicate what parameters are autocorrelated

**Authors response:** To address this comment we indicated the autocorrelated parameters "Although some of the surface variables were autocorrelated (e.g., TALK-salinity, pH-pCO$_2$ and nitrate-phosphate), …"

Figure 5: Fix the values label to tick marks, as well as, include in parenthesis the % variability explained by each RDA axes

**Authors response:** We fixed the tick marks and included the variability explained by the RDA (%) in the axis titles.

Line 392: A more recent study that Eppley et al., 1969 comparing the two species growth at different nutrient levels is: New Zealand Journal of Marine and Freshwater Research, 1995, Vol. 29: 345-357. Coccolithophores Gephyrocapsa oceanica and Emiliania huxleyi (Prymnesiophyceae = Haptophyceae) in New Zealand's coastal waters: characteristics of blooms and growth in laboratory culture.

**Authors response:** We included the suggested reference in line 392.

Lines 396-398: Elaborate better this sentence as sounds recursive or meaningless

**Authors response:** We decided to delete this sentence.

Line 408: The life cycle is not mention in Beaufort et al. 2011 study, but with species-environmental responses most related to shift in assemblage composition

**Authors response:** We changed "life cycle" to "physiological response".

Line 470: The statement seems correspond to Figure 4

**Authors response:** We fixed to "Figure 4".

Lines 478-479: Delete quotation marks

**Authors response:** Quotation marks were removed.

**Report 3**

I have noticed only two technical corrections to be fixed:

- Line 604: erase "314" from the reference

**Authors response:** The number was deleted.

- Line 677: correct the journal abbreviation by erasing "and". Use "Paleoceanogr. Paleoclimatology" instead.

**Authors response:** Fixed.

**Additional comments by the authors**

We fixed minor typos throughout the manuscript and named in lines 30, 217, 294, 300, 306, 383, 397, 398, and 524, the "ratio σ/ks vs. mean ks" as "normalized ks variation" as suggested in the other round of reviews to better characterize the new developed index.